# Nutritional Management of Functioning GEP-NENs

**DOI:** 10.3390/nu17132175

**Published:** 2025-06-30

**Authors:** Maribel del Olmo-García, Lorena Hernandez-Rienda, Maria Argente Pla, Juan F. Merino-Torres

**Affiliations:** 1Joint Research Unit on Endocrinology, Nutrition and Clinical Dietetics, Research Institute La Fe, University of Valencia-Health, 46026 Valencia, Spain; lorenahernandezrienda@gmail.com (L.H.-R.); argente_mar@gva.es (M.A.P.); merino_jfr@gva.es (J.F.M.-T.); 2Department of Medicine, Faculty of Medicine, University of Valencia, 46010 Valencia, Spain; 3Department of Endocrinology and Nutrition, University and Polytechnic Hospital La Fe, 46026 Valencia, Spain

**Keywords:** neuroendocrine tumors, malnutrition, carcinoid syndrome, VIPoma, glucagonoma, insulinoma, gastrinoma, somatostatinoma, nutritional support, GEP-NENs

## Abstract

Background: Functioning gastroenteropancreatic neuroendocrine neoplasms (GEP-NENs) are rare tumors that secrete biologically active hormones, leading to complex clinical syndromes such as carcinoid syndrome, VIPoma, glucagonoma, gastrinoma, insulinoma, and somatostatinoma. These syndromes frequently induce profound metabolic, gastrointestinal, and nutritional disturbances. Objective: This review aims to provide a comprehensive overview of the physiopathology of malnutrition in functioning GEP-NENs and to highlight nutritional and supportive care strategies, including how medical, surgical, and locoregional treatments can indirectly improve nutritional outcomes. Methods: We analyzed the current literature and clinical guidelines to identify key mechanisms of malnutrition across different functioning syndromes and their clinical manifestations. Nutritional recommendations and the impact of treatment modalities on nutritional status were summarized. Results: The pathophysiology of malnutrition in functioning NENs is multifactorial and syndrome-specific. Hormonal hypersecretion may cause diarrhea, electrolyte imbalances, catabolic states, steatorrhea, or hypoglycemia, among other effects. These lead to nutrient loss, malabsorption, or altered intake. Tailored dietary interventions, micronutrient supplementation (e.g., niacin, calcium, vitamin B12), and symptom-guided nutritional support are essential. Somatostatin analogs, PRRT, and cytoreductive approaches often contribute to symptom control, thereby enhancing nutritional status and patient quality of life. Conclusions: Malnutrition in functioning GEP-NENs is a significant clinical issue that requires early recognition and a multidisciplinary, individualized management plan. Integrating nutrition into the comprehensive care of these patients is essential to improve outcomes and quality of life.

## 1. Introduction

Neuroendocrine neoplasms (NENs) are a heterogeneous group of malignancies arising from neuroendocrine cells located in endocrine glands or from diffuse neuroendocrine cells present in organs such as the gastrointestinal tract, pancreas, or lungs. The incidence of these tumors has been increasing over the last few decades, reaching an age-adjusted incidence rate of 6.98 cases per 100,000 inhabitants in 2012, according to the Surveillance, Epidemiology, and End Results (SEER) program [1,2]. Among NENs, those originating from the gastroenteropancreatic (GEP) system represent the majority, accounting for over 60% of diagnosed cases [3].

The onset of NENs varies widely, with the highest incidence occurring in the sixth decade of life. However, in the context of hereditary syndromes, such as multiple endocrine neoplasia type 1 (MEN1) or Von Hippel–Lindau disease, the diagnosis can occur much earlier [4].

Patients with GEP-NENs often experience significant nutritional challenges due to a combination of tumor-related factors, treatment side effects, and metabolic disruptions. Both the tumor itself and the therapies used can directly affect the patient’s nutritional status, leading to malnutrition, sarcopenia, and vitamin or trace element deficiencies [5,6,7,8,9]. According to the Global Leadership Initiative on Malnutrition (GLIM), malnutrition is defined as a condition resulting from inadequate intake or assimilation of nutrients, characterized by both phenotypic and etiologic criteria. The diagnosis requires at least one phenotypic criterion (such as unintentional weight loss, low body mass index, or reduced muscle mass) and one etiologic criterion (such as reduced food intake/assimilation or the presence of disease burden/inflammation) [10].

Recent studies have highlighted the significant prevalence of nutritional disorders among patients with GEP-NENs. For instance, the NUTRIGETNE study reported that malnutrition in advanced GEP-NENs was prevalent in 61.9% of patients with advanced cancer, with low muscle mass being the most common criterion, affecting 50.9% of the cohort [2]. These findings underscore the critical need for comprehensive nutritional assessments and interventions in this patient population to improve clinical outcomes and quality of life.

Optimizing nutritional support can improve symptom control, enhance treatment tolerability, and ultimately contribute to better clinical outcomes in patients with these challenging malignancies. In line with ESPEN recommendations for oncology patients, a multimodal approach combining individualized dietary counseling, oral nutritional supplementation, and, when indicated, enteral or parenteral nutrition is essential to address malnutrition in this setting [11]. In selected cases of cancer-associated cachexia or refractory weight loss, emerging anabolic therapies, including selective androgen receptor modulators and investigational agents, such as ponsegromab (an anti-GDF15 monoclonal antibody targeting inflammation-driven anorexia and muscle wasting), offer promising avenues to improve nutritional status and preserve functional capacity [12].

Nutritional deterioration in patients with GEP-NENs arises from multiple factors; however, up to 30% of cases present with hormone-related syndromes at diagnosis, further exacerbating gastrointestinal and metabolic complications [13,14].

The aim of this review is to comprehensively address the nutritional management of patients with functioning GEP-NENs. By examining the complex interplay between tumor biology, treatment effects, and metabolic disturbances, this review highlights the critical need for early nutritional assessment and intervention. Optimizing nutritional support can improve symptom control, enhance treatment tolerability, and ultimately contribute to better clinical outcomes in patients with these challenging malignancies [6,7,15,16].

## 2. Carcinoid Syndrome

### 2.1. Physiopathology of Malnutrition

Carcinoid syndrome (CS) results from the excessive secretion of serotonin and other bioactive substances (e.g., histamine, kallikrein, bradykinin), most commonly by neuroendocrine tumors of the small intestine or lungs. These substances induce a range of systemic symptoms, including diarrhea, flushing, bronchospasm, carcinoid heart disease (CHD), mesenteric fibrosis, and niacin deficiency [14], each contributing to malnutrition through different mechanisms [15,16,17] (Figure 1).

### 2.2. Nutritional and Supportive Measures

Although dietary modifications are frequently recommended for symptom management in patients with neuroendocrine tumors (NETs) and carcinoid syndrome (CS), current scientific evidence supporting their effectiveness remains limited, and no major clinical guidelines offer standardized dietary protocols. Nevertheless, clinical practice suggests that small, frequent meals and the avoidance of aged cheeses, fermented foods, and other amine-rich products may help reduce symptoms in selected patients [6,17]. Given the heterogeneous presentation of CS, individualized nutritional management is essential (Table 1).

While dietary modifications and medical therapies are central to managing nutritional decline in carcinoid syndrome, they may be insufficient in advanced or complicated cases. In particular, patients with severe mesenteric fibrosis, bowel obstruction, or refractory diarrhea may develop functional intestinal failure, precluding adequate oral or enteral nutrition. In such scenarios, parenteral nutrition (PN) becomes necessary to maintain nutritional status and prevent further metabolic deterioration. Recent data indicate that home parenteral nutrition is a feasible option in carefully selected patients with NET-associated intestinal failure, including those with carcinoid syndrome, offering potential improvements in nutritional status and quality of life when managed by experienced multidisciplinary teams [18].

### 2.3. Medical Treatment

In patients with CS, nutritional support and dietary supplementation are key pillars of comprehensive care. However, certain medical treatments aimed at controlling hormonal hypersecretion also play a valuable indirect role in mitigating nutrition-related complications. Somatostatin analogs (SSAs), such as octreotide and lanreotide, are the first-line therapy in CS because of their ability to inhibit the secretion of serotonin and other bioactive substances. By reducing hormonal output, SSAs help alleviate symptoms such as diarrhea and flushing, thereby contributing to better nutrient absorption, reduced fluid and electrolyte losses, and improved dietary tolerance. In cases of severe or refractory symptoms, dose escalation or the use of short-acting formulations may be necessary to achieve symptom stabilization and prevent further nutritional decline [19,20,21].

For patients who remain symptomatic despite SSA therapy, telotristat ethyl, a tryptophan hydroxylase inhibitor, offers an additional therapeutic avenue. By directly reducing serotonin synthesis, telotristat has been shown to significantly decrease bowel frequency, particularly in patients with refractory diarrhea [22]. Adjunctive treatments, such as bile acid sequestrants, non-specific antidiarrheal agents (e.g., loperamide, diphenoxylate), and serotonin receptor antagonists (e.g., ondansetron), may also offer symptom relief [23]. Therefore, integrating pharmacologic therapies [24] with individualized nutritional strategies is essential to address both the underlying disease and its impact on the patient’s nutritional well-being (Table 2).

### 2.4. Surgical and Locoregional Therapies

Surgical debulking and locoregional liver-directed therapies can play a supportive role in improving the nutritional status of patients with carcinoid syndrome by rapidly reducing tumor burden and hormone secretion. Cytoreductive surgery, even when not curative, often leads to significant relief of symptoms, such as diarrhea and flushing, enabling better nutrient absorption and food tolerance. Similarly, liver-directed treatments, like TACE, TAE, SIRT, or RFA, help decrease hormonal output from hepatic metastases, enhancing symptom control and complementing medical therapies. Additionally, systemic approaches such as PRRT (e.g., Lu-177-DOTATATE) have been shown to improve functional status and quality of life, indirectly supporting nutritional recovery by stabilizing gastrointestinal symptoms and reducing the frequency of debilitating diarrhea [24,25,26].

## 3. VIPoma: Werner–Morrison Syndrome

### 3.1. Physiopathology of Malnutrition

VIPomas are rare functional neuroendocrine tumors characterized by the overproduction of vasoactive intestinal peptide (VIP), with an estimated incidence of 0.05 to 0.2 cases per million people per year, resulting in a distinct clinical syndrome often referred to as WDHA (watery diarrhea, hypokalemia, and achlorhydria). The most prominent manifestation is severe secretory diarrhea, which may reach volumes of 6 to 8 L per day, leading to profound fluid and electrolyte losses, particularly potassium and bicarbonate. In severe cases, these pathophysiological disturbances can result in malnutrition, cachexia, and the need for parenteral fluid and nutritional support. The nutritional consequences are further exacerbated by anorexia, nausea, and generalized weakness associated with chronic electrolyte imbalances. Therefore, early and aggressive management of VIPoma-related symptoms is essential not only for hormonal control but also to prevent or reverse the rapid decline in nutritional and functional status [27,28,29,30,31] (Figure 2).

### 3.2. Nutritional and Supportive Measures

In patients with VIPomas, the cornerstone of nutritional and clinical management is the rapid correction of fluid and electrolyte imbalances, as these disturbances are the main drivers of morbidity. The profound watery secretory diarrhea caused by excessive VIP secretion leads to massive losses of potassium, bicarbonate, sodium, and fluids, resulting in hypokalemia, metabolic acidosis, and severe dehydration. These imbalances compromise not only systemic homeostasis but also digestive function, nutrient absorption, and overall oral intake. In moderate to severe cases, intravenous fluid replacement therapy with normal saline, potassium chloride, and sodium bicarbonate is essential to restore equilibrium. Nutritionally, patients may require temporary bowel rest, oral rehydration solutions, or, in more critical scenarios, parenteral nutrition or electrolyte supplementation, particularly when oral intake is insufficient or poorly tolerated [16,27,29,31,32] (Table 3).

As diarrhea subsides, dietary intake can progressively be reintroduced, focusing on low-residue, low-lactose, and easily digestible foods, with close monitoring and repletion of potassium, bicarbonate, magnesium, and calcium levels. Early collaboration with nutrition support teams is critical to prevent malnutrition, promote recovery, and reduce the risk of complications such as cardiac arrhythmias, weakness, or renal dysfunction related to persistent electrolyte derangement.

### 3.3. Medical Treatment

Medical treatment plays a central role in improving the nutritional status of patients with VIPomas, primarily through a reduction in VIP secretion and the associated severe secretory diarrhea. Somatostatin analogs (SSAs) are the mainstay of therapy, achieving a reduction in diarrhea frequency and volume in approximately 65–85% of patients, which directly improves hydration, electrolyte balance, and gastrointestinal function—critical factors in preventing or reversing malnutrition. Therefore, maintenance of SSA therapy is strongly recommended even when other systemic treatments are introduced for tumor control. Among targeted therapies, sunitinib has shown variable but potentially high rates of symptomatic response (30–100%), while everolimus has demonstrated limited efficacy in VIPomas, with symptom control in fewer than 10% of cases. Peptide receptor radionuclide therapy (PRRT) with 177Lu-DOTATATE has shown a symptomatic response rate of around 80% and a disease control rate of 67%, supporting its role in reducing hormone-mediated symptoms and enabling better nutritional intake. Importantly, discontinuation of long-acting SSA therapy before PRRT has been associated with rapid recurrence of severe diarrhea, warranting the use of short-acting octreotide up to the time of radiopharmaceutical administration. Cytotoxic chemotherapy remains effective in many cases, with reports of both tumor and symptom control in patients with VIPoma. Additionally, glucocorticoids and interferon-alpha have been reported to improve hormonal control in select cases. By reducing symptom burden, these therapies not only help stabilize the disease but also create the necessary conditions for adequate nutrition, recovery of body weight, and prevention of further metabolic complications [13,28,29,30].

### 3.4. Locoregional and Surgical Therapies

Following initial clinical stabilization, surgical resection should be pursued in patients with locally confined VIPomas, as it represents the only potentially curative treatment. Complete removal of the tumor eliminates the source of vasoactive intestinal peptide (VIP), resulting in the resolution of secretory diarrhea, restoration of fluid and electrolyte balance, and, ultimately, the normalization of gastrointestinal function. These effects directly enhance nutritional recovery, allowing for improved oral intake, nutrient absorption, and reversal of catabolic states.

In patients with unresectable metastatic disease, particularly those with dominant hepatic involvement, the therapeutic goal shifts toward tumor burden reduction and hormonal symptom control. Debulking surgery can significantly reduce VIP secretion by removing a large portion of the tumor mass, which helps alleviate persistent diarrhea and minimizes fluid and nutrient losses. Similarly, liver-directed therapies, such as transarterial embolization (TAE), radioembolization (SIRT), radiofrequency or microwave ablation, and cryoablation, have been used successfully in selected patients to control hormone output and stabilize symptoms. While not curative, these interventions support nutritional rehabilitation by improving hydration status, reducing electrolyte imbalances, and enhancing tolerance to oral or enteral feeding. In this context, integrating locoregional control strategies with medical and nutritional therapies is essential to achieving both symptom relief and metabolic recovery [4,13,30,33].

## 4. Glucagonoma

### 4.1. Physiopathology of Malnutrition

Glucagonomas are extremely rare functional pancreatic neuroendocrine tumors (pNETs), with an estimated incidence of 0.01 to 0.1 new cases per 10^6^ population per year [33]. These tumors are characterized by the excessive secretion of glucagon, leading to a distinct clinical syndrome with multiple systemic manifestations that significantly impair nutritional status. Among the hallmark features is necrolytic migratory erythema, a painful dermatological condition often associated with zinc deficiency, and frequently accompanied by glossitis, angular cheilitis, and stomatitis, all of which can limit oral intake due to pain or mucosal irritation [32,34,35]. Additional findings, such as normocytic normochromic anemia, deep vein thrombosis, depression, and mild diabetes mellitus, further complicate the clinical picture and may contribute to anorexia, systemic inflammation, and catabolic stress.

Importantly, glucagon itself exerts a strong catabolic effect, promoting hepatic gluconeogenesis and lipolysis while reducing protein synthesis and increasing proteolysis. This results in a state of hypoaminoacidemia, which impairs tissue repair and contributes to muscle wasting. When compounded by diarrhea, a frequent symptom in glucagonoma, the patient may experience profound fluid, electrolyte, and nutrient losses [34,35]. The combination of catabolism, reduced intake due to mucocutaneous lesions, and digestive losses often culminates in rapid and significant weight loss, placing the patient at high risk for severe malnutrition and cachexia. Early nutritional assessment and support are therefore essential components of glucagonoma management (Figure 3).

### 4.2. Nutritional and Supportive Measures

Nutritional support in patients with glucagonoma must be initiated early, ideally at diagnosis, and before the implementation of any surgical or systemic therapy. Management should begin with general dietary strategies for diarrhea, complemented by symptom-specific interventions tailored to the functional hormonal syndrome. The catabolic effects of glucagon, which drive accelerated protein breakdown, gluconeogenesis, and lipolysis, often result in severe and rapid weight loss, necessitating intensive nutritional support [6,16,32,34,35] (Table 4).

### 4.3. Medical Treatment

Medical treatment in glucagonoma plays a pivotal role in improving the patient’s nutritional status by reducing the tumor’s hormonal output and controlling symptoms that directly impair nutrient intake and absorption. Somatostatin analogs (SSAs) are considered first-line therapy, effectively decreasing glucagon secretion and alleviating hallmark symptoms, such as diarrhea and necrolytic migratory erythema, thereby improving nutrient retention, skin integrity, and oral intake [6,13,34]. Peptide receptor radionuclide therapy (PRRT) with 177Lu-DOTATATE has shown a high rate of symptomatic response (71%) and improved quality of life, indirectly facilitating nutritional recovery by stabilizing metabolic disturbances and reducing catabolic stress [13,16]. While targeted therapies, such as everolimus and sunitinib, have shown some biochemical or radiologic activity in pancreatic NETs, their specific role in glucagonoma remains uncertain and potentially limited, especially given everolimus’ potential to worsen glycemic control. Cytotoxic chemotherapy, including streptozotocin with 5-FU or temozolomide-based regimens, has demonstrated clinical efficacy in a subset of patients and may support symptom reduction and nutritional stabilization in progressive or metastatic disease [21,36,37]. Ultimately, successful medical management of glucagonoma reduces the burden of catabolic and mucocutaneous symptoms, creating a window for effective nutritional rehabilitation [4,13,33].

### 4.4. Surgical and Locoregional Treatments

In patients with glucagonoma, surgical resection remains the only potentially curative treatment and is the primary goal when the disease is localized and technically resectable. Complete tumor removal results in the rapid cessation of glucagon hypersecretion, leading to the resolution of diarrhea, an improvement in mucocutaneous symptoms, such as necrolytic migratory erythema, and the reversal of catabolic processes. These changes collectively restore the patient’s ability to maintain adequate oral intake and improve nutrient absorption, thereby facilitating nutritional recovery. In cases of unresectable or metastatic disease, cytoreductive surgery (debulking) or liver-directed locoregional therapies—such as embolization, radioembolization, or ablation—may reduce tumor burden and glucagon secretion. Although not curative, these approaches can contribute significantly to symptom control, improving quality of life and creating a more favorable metabolic environment for nutritional support and rehabilitation. Thus, integrating surgical or locoregional strategies into the treatment plan not only targets tumor control but also alleviates the hormonal and catabolic complications that drive malnutrition in glucagonoma [4,13,33].

## 5. Gastrinoma: Zollinger–Ellison Syndrome

### 5.1. Physiopathology of Malnutrition

Gastrinomas are rare functional neuroendocrine tumors, most commonly arising in the pancreas or duodenum, and are typically associated with Zollinger–Ellison syndrome (ZES), a clinical condition resulting from excessive secretion of gastrin. This hormone overstimulates the gastric parietal cells, leading to marked hypersecretion of hydrochloric acid, which profoundly alters the gastrointestinal environment and digestion [1,4,38,39] (Figure 4).

### 5.2. Nutritional and Supportive Measures

Nutritional support in patients with gastrinoma and Zollinger–Ellison syndrome (ZES) is centered on mitigating the effects of chronic gastric acid hypersecretion, protecting the gastrointestinal mucosa, and correcting associated nutritional deficiencies (Table 5).

Maintaining adequate hydration is essential to support mucosal integrity and prevent complications, such as dehydration or nephrolithiasis, which are more common in ZES because of excessive acid load.

Pharmacological management plays a key supportive role, particularly with the long-term use of high-dose proton pump inhibitors (PPIs) to suppress acid secretion and control ulcerative disease. However, chronic PPI therapy is associated with impaired absorption of calcium and vitamin B12, necessitating proactive supplementation. Calcium citrate is preferred because of its superior absorption, independent of gastric pH, and should be taken separately from PPIs to optimize uptake. Vitamin B12 deficiency may develop over time, requiring supplementation either orally or via intramuscular injection to prevent hematological and neurological complications. Additionally, iron supplementation may be necessary in cases of chronic gastrointestinal bleeding leading to iron-deficiency anemia. In patients who develop steatorrhea, possibly due to bile salt malabsorption or pancreatic enzyme inactivation, tailored dietary and enzymatic interventions are warranted [6,32,38,39].

### 5.3. Medical Treatment

Medical therapies in gastrinoma are primarily aimed at reducing gastrin hypersecretion and controlling tumor growth, which, in turn, can lead to significant improvements in nutritional status by mitigating the downstream effects of acid hyperproduction, including ulceration, malabsorption, and diarrhea. Somatostatin analogs (SSAs) have demonstrated efficacy in numerous studies, including randomized and non-randomized trials, particularly in pancreatic and functional NETs such as gastrinomas [4,19,38,39]. In one cohort, 67% of patients with Zollinger–Ellison syndrome (ZES) showed sustained biochemical control with SSAs, while others required dose escalation to maintain therapeutic benefit [38,39]. By reducing gastrin levels and acid secretion, SSAs can directly improve nutrient absorption, reduce mucosal injury, and allow better tolerance of oral intake.

Targeted therapies, such as sunitinib and everolimus, though not specifically studied in isolated gastrinoma populations, have demonstrated progression-free survival benefits in broader cohorts of pancreatic NETs [20,21]. While their direct impact on gastrin levels remains unclear, tumor control may indirectly help reduce hormonal output in certain cases. Peptide receptor radionuclide therapy (PRRT) has emerged as a valuable option for refractory hormone-related symptoms, including in ZES. In a subset of metastatic gastrinoma patients treated with 90Y and/or 177Lu-based therapies, studies reported reductions in serum gastrin levels by up to 81%, along with tumor response in over 50% of cases. Although recurrence rates remain high, PRRT may offer a critical therapeutic window for symptom relief and nutritional stabilization in patients resistant to standard interventions [40]. Chemotherapy (e.g., temozolomide and capecitabine) is generally reserved for advanced or progressive disease but may be considered in selected cases to reduce tumor burden and improve clinical and nutritional outcomes when other therapies have failed [6,38].

### 5.4. Surgical and Locoregional Treatments

Surgical and locoregional therapies play a pivotal role in the comprehensive management of gastrinoma, not only for disease control but also for their potential to improve nutritional outcomes by reducing hormone-related gastrointestinal complications. In accordance with current clinical guidelines and expert consensus, surgical resection should be considered in all patients with localized gastrinoma where the anticipated benefit outweighs the surgical risk. Curative surgery, when feasible, can eliminate the source of gastrin hypersecretion, effectively resolving acid-related complications, such as refractory peptic ulcers, diarrhea, and malabsorption, and thereby enabling full nutritional rehabilitation. Even in the setting of advanced or metastatic disease, cytoreductive surgery (debulking) may be considered when ≥90% of the tumor burden can be resected. Although not supported by randomized trials, this strategy has been associated with symptom improvement and stabilization of hormonal excess, which can translate into better nutrient absorption and weight recovery.

In patients with unresectable metastatic disease, various locoregional liver-directed therapies, including radiofrequency ablation, ethanol injection, cryotherapy, and transarterial embolization (TAE), chemoembolization (TACE), or radioembolization, may be employed to reduce hepatic tumor burden and lower circulating gastrin levels. While these approaches are palliative, they can significantly alleviate gastrin-driven symptoms and reduce the need for high-dose medical therapy, thereby contributing to the restoration of gastrointestinal function and improved nutritional status. When integrated with ongoing dietary support and pharmacologic acid suppression, these interventions help create a more stable metabolic environment for effective nutritional recovery in patients with gastrinoma [4,13,33].

## 6. Somatostatinoma

### 6.1. Physiopathology of Malnutrition

Somatostatinomas are rare neuroendocrine tumors that secrete excessive amounts of somatostatin, though overt hormonal syndromes are clinically evident in fewer than 5% of cases. When present, the somatostatinoma syndrome typically includes a constellation of symptoms, such as diabetes mellitus, diarrhea or steatorrhea, cholelithiasis, hypochlorhydria, and weight loss, each of which contributes to a profound risk of malnutrition [13,16] (Figure 5).

Unlike some other functional NETs, the nutritional compromise in somatostatinoma is typically multifactorial, resulting not only from hormone-related gastrointestinal losses but also from pancreatic insufficiency, biliary complications, and disordered glucose metabolism. Consequently, these patients often experience both quantitative and qualitative malnutrition, which may be underrecognized if only body weight is assessed. This complex and overlapping clinical picture necessitates a multimodal management approach that integrates endocrine control, exocrine pancreatic support, and targeted nutritional strategies to prevent or reverse malnutrition and improve overall outcomes [13,16,32].

### 6.2. Nutritional Intervention and Support

Nutritional management in patients with somatostatinoma requires a multifaceted approach tailored to the specific gastrointestinal and metabolic disturbances caused by somatostatin excess. One of the central manifestations is steatorrhea, resulting from impaired pancreatic exocrine function and fat malabsorption (Table 6).

Weight loss is another critical concern in somatostatinoma, particularly when compounded by diabetes and persistent diarrhea. Close monitoring of nutritional status, body composition, and micronutrient levels is essential, and early involvement of a clinical nutrition team can significantly improve outcomes [6,7,16,32].

### 6.3. Medical Treatment

Medical management in somatostatinoma focuses on controlling hormonal hypersecretion and tumor growth, both of which are directly linked to the nutritional deterioration observed in these patients. Somatostatin analogs (SSAs) are the treatment of choice and represent the mainstay of therapy, particularly in functional neuroendocrine tumors (NETs) that cause peptide-related syndromes. In somatostatinoma, SSAs have been shown to provide significant symptomatic relief, notably in reducing diarrhea and weight loss, thereby improving nutrient retention, absorption, and overall metabolic stability. By inhibiting the secretion of somatostatin and its downstream effects on pancreatic enzyme suppression and gastrointestinal motility, SSAs contribute to better control of steatorrhea and malabsorption, ultimately facilitating more effective nutritional support [13,41].

Among other systemic therapies, the evidence remains limited. Peptide receptor radionuclide therapy (PRRT) with 177Lu-DOTATATE has shown clinical benefit in patients with hormone-secreting NETs by improving refractory hormonal syndromes; however, specific data on its efficacy in metastatic somatostatinoma are currently lacking. Similarly, everolimus, despite its antiproliferative activity in other pNETs, may worsen glucose metabolism by reducing insulin secretion and promoting insulin resistance—effects that could further compromise nutritional status, particularly in patients already experiencing diabetes mellitus. Sunitinib and cytotoxic chemotherapy have not yet demonstrated consistent efficacy in controlling the hormonal symptoms specific to somatostatinoma, and no data are currently available regarding their nutritional impact in this context. Therefore, while SSA therapy remains the most effective and reliable approach to stabilizing both hormonal activity and nutritional decline, additional evidence is needed to clarify the role of other systemic treatments in the supportive care of patients with somatostatinoma [4,13,33,41].

### 6.4. Surgical and Locoregional Therapies

As with other functioning neuroendocrine tumors (NETs), surgical resection remains the only potentially curative treatment for somatostatinoma and offers the most definitive strategy for controlling hormone-related symptoms. Complete tumor removal results in the normalization of somatostatin levels, thereby alleviating its broad inhibitory effects on gastrointestinal and pancreatic functions. This hormonal correction can lead to the resolution of steatorrhea, malabsorption, and diabetes-related metabolic instability, ultimately allowing for the restoration of nutritional status. In cases of unresectable or metastatic disease, treatment focuses on tumor stabilization and palliation of symptoms by reducing somatostatin secretion. Cytoreductive (debulking) surgery, though not curative, may reduce tumor burden sufficiently to provide symptomatic and nutritional benefit [4,13,33].

Locoregional liver-directed therapies, including radioembolization, radiofrequency ablation, microwave ablation, and cryoablation, can also be considered in patients with hepatic metastases, depending on the center’s experience and resource availability. These interventions aim to reduce hormone production and tumor volume, often enhancing the response to systemic therapy and helping to stabilize gastrointestinal function. By alleviating the hormonal drivers of diarrhea, enzyme deficiency, and catabolic stress, these procedures support the recovery of nutrient absorption and help reverse or prevent severe malnutrition. Therefore, surgical and locoregional interventions play an important complementary role in the nutritional rehabilitation of patients with advanced somatostatinoma [4,13,33].

## 7. Insulinoma

### 7.1. Physiopathology of Malnutrition

Insulinomas are rare pancreatic neuroendocrine tumors (pNETs), with an estimated incidence of one to three cases per million people annually. Most are benign and slow-growing, but their clinical significance stems from their inappropriate secretion of insulin, which leads to recurrent episodes of hypoglycemia. These episodes are classically described by the Whipple triad: (1) symptoms of hypoglycemia, (2) documented low plasma glucose levels during symptoms, and (3) relief of symptoms following normalization of blood glucose. These frequent and often unpredictable episodes cause significant dietary disruption to relieve or prevent hypoglycemic symptoms. Contrary to other functional neuroendocrine tumors, insulinoma-related malnutrition is not always associated with weight loss. In fact, a subset of patients experience progressive weight gain. This weight gain may mask underlying metabolic imbalance, increased fat mass at the expense of lean tissue, poor dietary quality, and erratic glycemic control [8,42,43] (Figure 6).

### 7.2. Nutritional Management and Support

Nutritional intervention is the cornerstone of medical management in insulinoma and other forms of endogenous hyperinsulinism. The primary objective is to stabilize glycemic fluctuations and prevent hypoglycemic episodes through strategic dietary modifications. In cases of severe or refractory hypoglycemia, more aggressive support may be needed [8,13,42,43,44,45] (Table 7).

Additionally, the use of continuous glucose monitoring (CGM), a technology widely validated in type 1 diabetes, has shown promise in insulinoma patients. CGM enables the early detection of hypoglycemia by providing real-time glucose trends, allowing for prompt dietary or medical adjustments. Beyond acute management, CGM supports pattern recognition, therapy response monitoring, and most importantly, patient education and empowerment, encouraging proactive engagement in nutritional self-management. Together, dietary strategies and emerging glucose-monitoring technologies are critical tools in ensuring metabolic stability and protecting the nutritional well-being of patients with insulinoma [8,44,45].

### 7.3. Medical Treatment

While dietary intervention is central to managing insulinoma-related hypoglycemia, it is often insufficient on its own, particularly in patients with severe or refractory hypoglycemia, necessitating the use of targeted medical therapies (Table 8).

In patients with unresectable metastatic disease, the therapeutic goal shifts toward tumor stabilization and hormonal suppression, always in conjunction with ongoing nutritional support. As with other functional neuroendocrine tumors, tumor-directed therapies, including peptide receptor radionuclide therapy (PRRT), chemotherapy, everolimus, and sunitinib, can indirectly improve nutritional outcomes by controlling insulin hypersecretion and reducing the frequency of hypoglycemia [4,13,33,42,43].

### 7.4. Surgical and Locoregional Therapies

In patients with insulinoma, definitive local therapies, such as surgery and locoregional interventions, play a pivotal role in stabilizing metabolism and nutrition by controlling the source of excess insulin secretion. For localized disease, surgical resection with curative intent is the first-line approach and often leads to long-term remission, as removing the insulin-secreting tumor eliminates hyperinsulinemia and promptly corrects recurrent hypoglycemia. This restoration of euglycemia allows patients to resume normal dietary intake without constant glucose supplementation, thereby improving nutritional status and preventing weight fluctuations. In selected cases of insulinoma, loco-regional therapies provide an effective, organ-preserving alternative for symptom control, often avoiding the need for surgery. Among these, Endoscopic Ultrasound (EUS)-guided radiofrequency ablation has shown promising results in achieving hormonal control and tumor reduction, particularly in high-risk surgical candidates [46]. In metastatic or unresectable cases, cytoreductive strategies, including debulking surgery and liver-directed therapies like transarterial embolization or ablation, are employed to reduce tumor burden and insulin output. While these interventions are palliative rather than curative, achieving a substantial reduction (often >70–90%) in tumor mass is associated with significantly better glycemic control and relief of hypoglycemic symptoms. Published guidelines and studies underscore that debulking an advanced insulinoma can “contribute to improved glycemic control”, which not only alleviates life-threatening hypoglycemia but also facilitates nutritional recovery by breaking the cycle of frequent feeding to counteract low blood sugar. Notably, liver-directed locoregional therapies can lead to high rates of symptomatic improvement (reported in ~70–100% of patients), reflecting effective hormone stabilization in cases of hepatic metastases. Thus, both curative surgical resection and adjunct locoregional treatments support improved metabolic and nutritional outcomes in insulinoma patients by normalizing insulin levels, which, in turn, restores glucose homeostasis and allows for balanced nutrition [4,13,33,42,43].

## 8. Conclusions

Malnutrition in patients with functioning gastroenteropancreatic neuroendocrine neoplasms is a multifactorial and frequently underestimated complication that significantly impacts prognosis, quality of life, and therapeutic response. Each functional syndrome, whether carcinoid syndrome, VIPoma, glucagonoma, gastrinoma, insulinoma, or somatostatinoma, presents with a distinct constellation of hormone-related metabolic and gastrointestinal disturbances, which translates into unique nutritional challenges. From debilitating diarrhea and electrolyte loss in carcinoid syndrome and VIPoma, to the catabolic weight loss of glucagonoma, the acid-related malabsorption of gastrinoma, the paradoxical weight gain from reactive hyperphagia in insulinoma, and the complex exocrine insufficiency in somatostatinoma, the pathophysiology of malnutrition is as diverse as it is profound.

What unites these syndromes is that their nutritional deterioration stems not only from inadequate intake or malabsorption but also from hormonal dysregulation, metabolic derangement, and treatment-related toxicity. As such, nutritional support is not ancillary, but essential, and should be implemented from the time of diagnosis, not merely as a reactive measure in advanced disease. Nutritional interventions must be personalized, proactive, and dynamic, evolving in tandem with the disease course and therapeutic strategy.

Crucially, medical therapies, such as somatostatin analogs, telotristat, diazoxide, and PRRT, not only alleviate symptoms but contribute secondarily to nutritional improvement by reducing hormone secretion, stabilizing metabolism, and controlling diarrhea or hypoglycemia. Similarly, surgical resection and liver-directed therapies, even when not curative, offer the potential for functional hormone suppression, which can break the cycle of malabsorption, catabolism, and micronutrient deficiencies.

A truly effective approach to managing these patients requires integrated, multidisciplinary care, where oncology, endocrinology, gastroenterology, surgery, and clinical nutrition collaborate seamlessly. Regular assessment of nutritional status, including body composition, micronutrient levels, and functional outcomes, should become a routine part of the care pathway. Nutritional therapy should not be viewed as supportive alone but as a therapeutic axis in its own right, capable of influencing clinical outcomes, patient resilience, and long-term survival.

In conclusion, nutrition in functioning GEP-NENs is both a clinical target and a therapeutic opportunity. Understanding the syndrome-specific mechanisms of malnutrition and tailoring interventions accordingly can transform the trajectory of care, from simply treating disease to truly supporting the person living with it.

## Figures and Tables

**Figure 1 nutrients-17-02175-f001:**
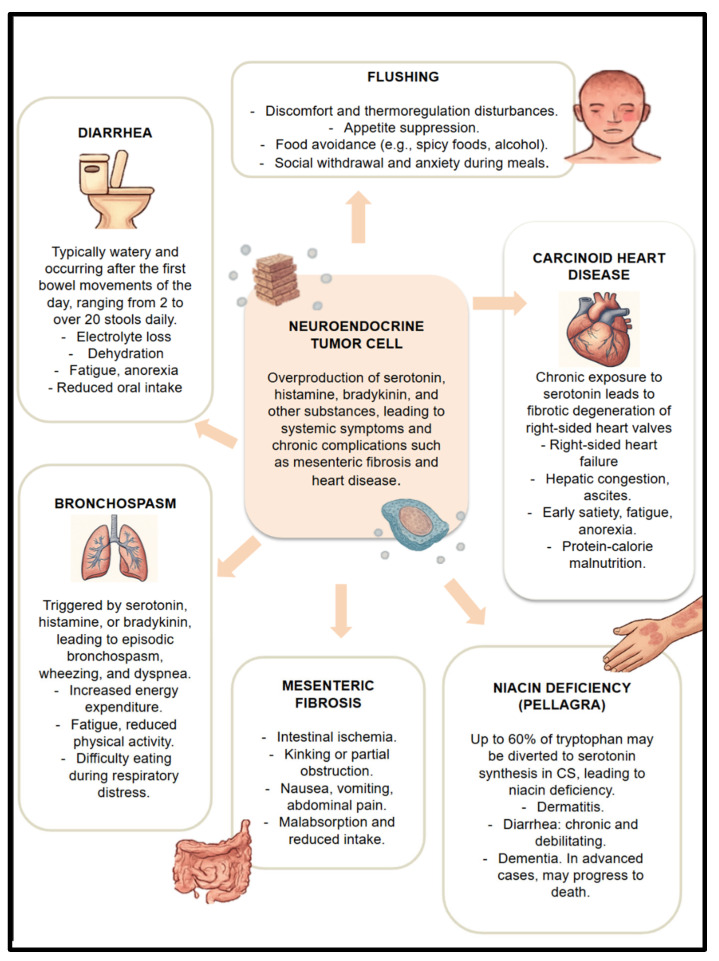
Physiopathology of malnutrition due to CS.

**Figure 2 nutrients-17-02175-f002:**
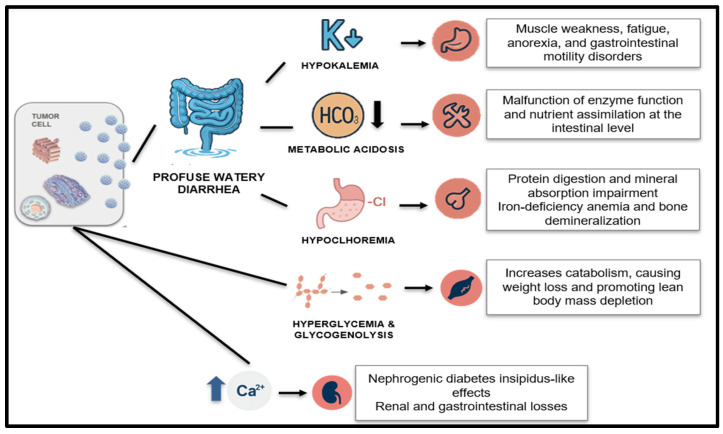
Physiopathology of malnutrition due to VIP secretion.

**Figure 3 nutrients-17-02175-f003:**
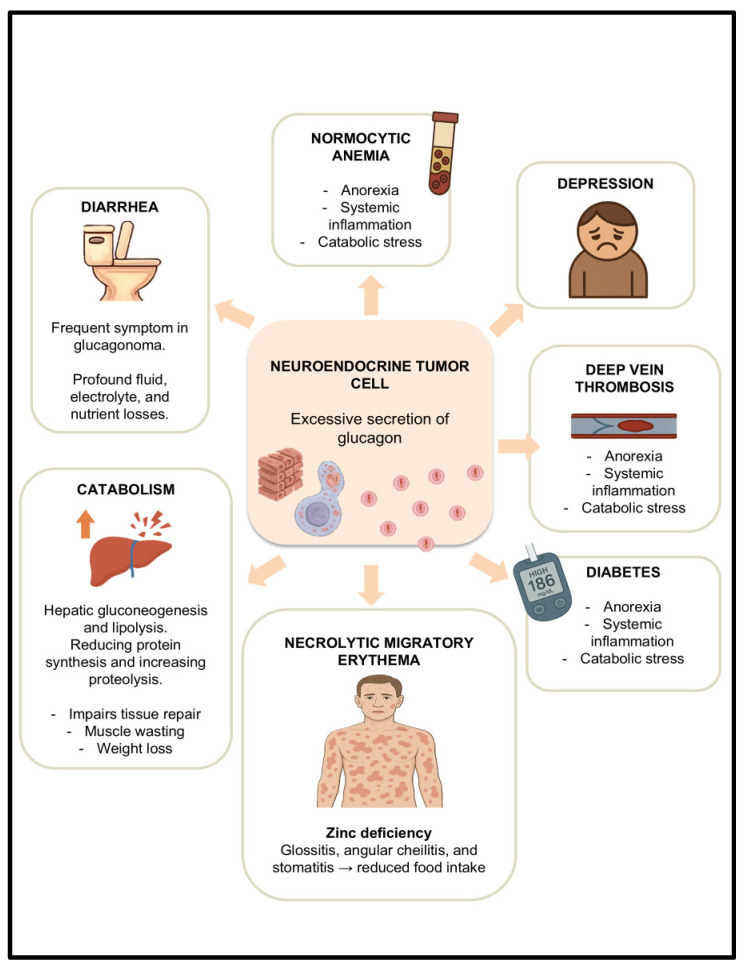
Physiopathology of malnutrition due to glucagon secretion.

**Figure 4 nutrients-17-02175-f004:**
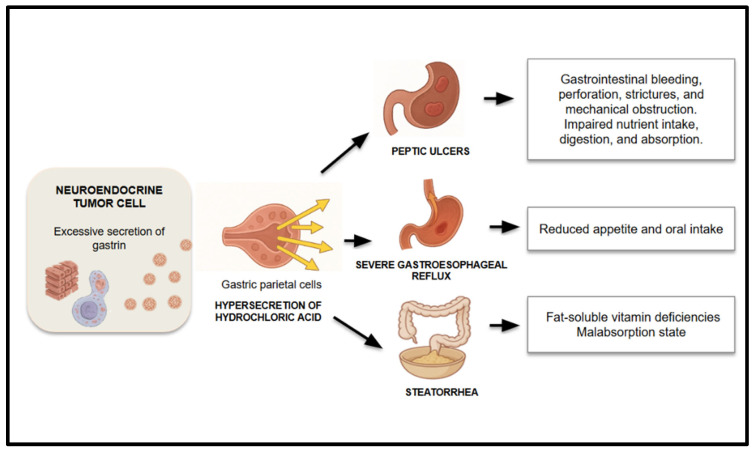
Physiopathology of malnutrition due to gastrin secretion.

**Figure 5 nutrients-17-02175-f005:**
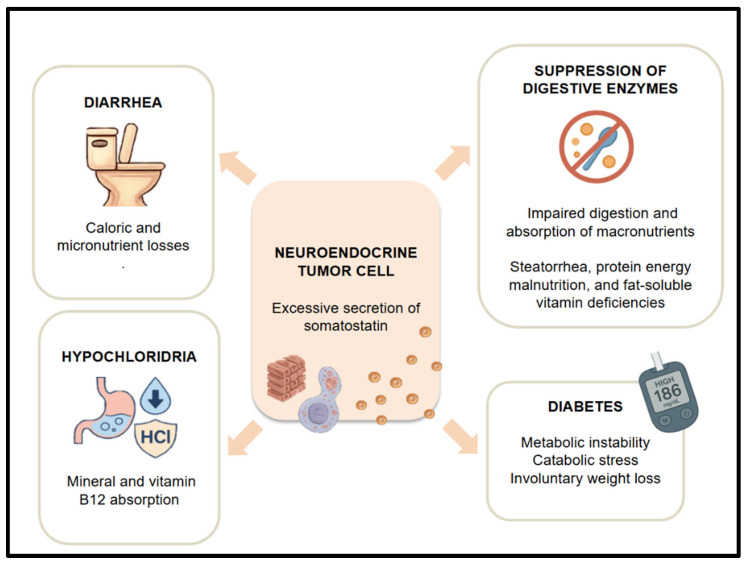
Physiopathology of malnutrition due to somatostatin secretion.

**Figure 6 nutrients-17-02175-f006:**
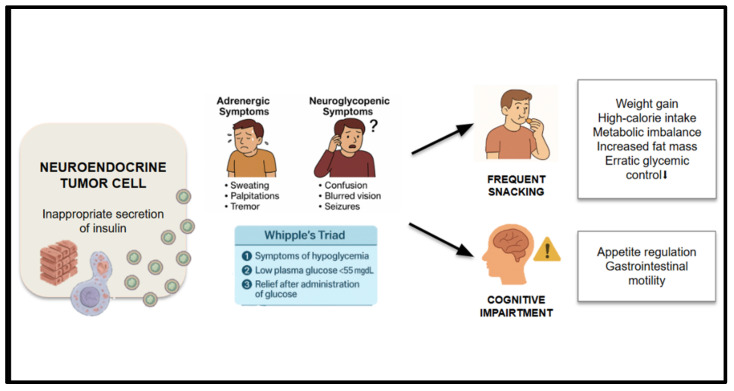
Physiopathology of malnutrition due to insulin secretion.

**Table 1 nutrients-17-02175-t001:** General nutritional and supportive measures for CS.

Symptoms	Recommendations
Diarrhea	Small, frequent meals; low-fiber diet; maintain adequate hydration; avoid only individual symptom-triggering foods.
Flushing	Avoid common triggers (e.g., alcohol, spicy foods, hot beverages); consider cooling foods; maintain hydration; individualized dietary monitoring to identify flushing triggers.
Carcinoid Heart Disease	Energy-dense, easy-to-digest meals; manage early satiety; monitor for signs of right-sided heart failure; sodium restriction may be necessary if fluid overload occurs.
Bronchospasm	Avoid large meals that can exacerbate breathing discomfort; ensure adequate calorie intake to offset increased energy expenditure from labored breathing.
Mesenteric Fibrosis	Soft, low-residue diet to minimize mechanical irritation; manage pain and bowel dysfunction; consider enzymatic support and monitor for obstruction-related symptoms.
Niacin Deficiency	Supplement with oral nicotinamide: 20–40 mg/day as prevention; therapeutic dose 100 mg every 6 h initially, then 50 mg every 8 h after improvement; some cases may require 500 mg/day [13].

**Table 2 nutrients-17-02175-t002:** Medical treatments and their nutritional impact in carcinoid syndrome.

Medical Treatment	Primary Symptom Target	Impact on Nutritional Status
Somatostatin Analogs (SSAs)	Hormonal hypersecretion (serotonin and others)	Reduces diarrhea and flushing; improves nutrient absorption, hydration, and tolerance to food.
Telotristat Ethyl	Serotonin synthesis (diarrhea control)	Reduces serotonin-driven diarrhea; improves hydration and overall gut function.
Bile Acid Sequestrants	Bile acid-induced diarrhea	Improves diarrhea; reduces fluid and electrolyte loss.
Loperamide/Diphenoxylate	Non-specific diarrhea	May reduce stool frequency, improving fluid retention; no direct effect on absorption.
Serotonin Receptor Antagonists (Ondansetron)	Diarrhea via serotonin pathway	May offer symptom relief in selected cases, indirectly supporting improved intake.
Cyproheptadine	Serotonin-related diarrhea	Limited use due to side effects; can reduce symptoms, allowing better nutritional intake in select cases.

**Table 3 nutrients-17-02175-t003:** General nutritional and supportive measures for VIPOMA.

Symptoms	Recommendations
Profuse watery diarrhea	Initiate oral rehydration if possible; in severe cases, IV hydration with close fluid balance monitoring.
Electrolyte imbalance (hypokalemia, bicarbonate loss)	Aggressive repletion of potassium and bicarbonate; dietary counseling to increase potassium-rich foods when tolerated.
Dehydration	IV fluids with electrolytes (saline, potassium, bicarbonate); monitor for ongoing losses.
Metabolic acidosis	Correct acidosis via IV bicarbonate; ensure adequate nutrition to avoid further catabolism.
Weight loss/malnutrition	High-calorie, nutrient-dense oral supplements; consider enteral nutrition; TPN in severe refractory cases.

**Table 4 nutrients-17-02175-t004:** General nutritional and supportive measures for glucagonoma.

Symptoms	Recommendations
Diarrhea	Maintain hydration (oral or IV), low-fiber diet, avoid lactose and high-fat foods, consider use of antidiarrheals if appropriate.
Painful glossitis, cheilitis, angular stomatitis	Soft or liquid diet (e.g., mashed potatoes, soft breads, soups); avoid spicy/citrus foods; oral rinses; good oral hygiene.
Normocytic anemia	Iron-, folate-, and vitamin B12-rich diet; consider red blood cell transfusion in severe cases.
Hypoalbuminemia/hypoproteinemia	Amino acid and albumin infusions as needed to correct deficits.
Zinc deficiency	Oral or IV zinc supplementation based on lab findings.
Necrolytic migratory erythema (NME)	Zinc and amino acid supplementation; corticosteroids and antibiotics if indicated.
Catabolic syndrome with significant weight loss	Essential fatty acid-rich oral supplements; initiate enteral nutrition; consider TPN in severe malabsorption cases.

**Table 5 nutrients-17-02175-t005:** General nutritional and supportive measures for gastrinoma.

Symptoms	Recommendations
Recurrent peptic ulcers/abdominal pain	Small, frequent meals; avoid large meals that stimulate acid secretion.
Gastroesophageal reflux disease (GERD)	Avoid acidic, spicy, and high-fat foods; maintain upright position after meals.
Chronic diarrhea ± steatorrhea	Low-fat diet; consider pancreatic enzyme replacement if steatorrhea is present.
Calcium and vitamin D malabsorption	Supplement with calcium (preferably citrate) and vitamin D; monitor bone density.
Vitamin B12 deficiency	Supplement with oral or intramuscular vitamin B12; monitor serum levels.
Iron-deficiency anemia	Iron supplementation (oral or IV depending on severity); treat underlying bleeding if present.
Hypergastrinemia-induced acid hypersecretion	Chronic high-dose proton pump inhibitor therapy (e.g., omeprazole 60–120 mg/day or equivalent); monitor for long-term complications, such as micronutrient malabsorption.

**Table 6 nutrients-17-02175-t006:** General nutritional and supportive measures for somatostatinoma.

Symptoms	Recommendations
Steatorrhea	Low-fat diet; small frequent meals; assess need for pancreatic enzyme replacement therapy (PERT).
Diarrhea	Hydration; low-residue diet; avoid lactose and high-fat foods; consider antidiarrheals.
Fat-soluble vitamin deficiencies (A, D, E, K)	Supplement vitamins A, D, E, and K orally or parenterally depending on severity.
Weight loss/malnutrition	Use oral nutritional supplements; consider enteral nutrition; escalate to parenteral nutrition if severe malabsorption occurs.

**Table 7 nutrients-17-02175-t007:** General nutritional and supportive measures for insulinoma.

Symptoms	Recommendations
Recurrent hypoglycemia	Frequent small meals every 2–3 h; include a bedtime snack.
Acute episodes of hypoglycemia	Rapid-acting carbohydrates (e.g., fruit juice with added glucose or sucrose).
Weight gain due to reactive eating	Emphasize low-glycemic index; slow-absorbing carbohydrates (e.g., whole grains, legumes).
Severe or refractory hypoglycemia	Consider continuous nasogastric feeding or IV glucose infusion in acute settings.
Glycemic instability	Use continuous glucose monitoring (CGM) to detect early hypoglycemia, guide dietary interventions, and empower patient self-management.

**Table 8 nutrients-17-02175-t008:** Medical treatments and their nutritional impact on insulinoma.

Medical Treatment	Nutritional Impact
Diazoxide	Raises blood glucose by inhibiting insulin secretion; allows reduced carbohydrate intake and stabilizes glycemia but may cause sodium retention and edema.
SSAs	Controls insulin secretion; improves fasting tolerance and reduces need for constant food intake but may impair counter-regulatory glucagon response.
Pasireotide	May help in refractory hypoglycemia; more potent inhibition of insulin but potential for hyperglycemia.
Calcium Channel Blockers (Verapamil, Diltiazem)	Variable effect; may inhibit insulin secretion in some cases; modest nutritional benefit.
Other Agents (Beta-blockers, Phenytoin, Corticosteroids)	Can support glycemia by enhancing gluconeogenesis and insulin resistance; generally used adjunctively.
PRRT (177Lu-DOTATATE)	Reduces tumor burden and hormone secretion in advanced cases; improves glucose stability and reduces risk of nutritional deterioration from hypoglycemia.
Chemotherapy	Helps control tumor progression in malignant insulinoma; supports overall metabolic control and nutritional recovery.
Everolimus/Sunitinib	In advanced disease, may stabilize glucose levels indirectly through tumor control; risk of side effects that could impair appetite or GI function;Everolimus raises glycemia as a secondary effect.

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
