# Peer review of "Nutritional Management of Functioning GEP-NENs"

_nutrients, 2025, doi:10.3390/nu17132175_

Round 1
Reviewer 1 Report
Comments and Suggestions for Authors
The authors have prepared a review on GNETS that is well structured, clear and comprehensive. The illustrations are very helpful as well.
I believe that a paragraph focused on medical nutrition therapy (mostly parenteral nutrition) could be useful. GNET associated intestinal failure is not rare and can be a major challenge as malnutrition can be very difficult to mitigate in these cases. There are some publications already reporting on this situation.(Feasibility of Home Parenteral Nutrition in Patients with Intestinal Failure Due to Neuroendocrine Tumours: A Systematic Review.Clement DSVM, Brown SE, Naghibi M, Cooper SC, Tesselaar MET, van Leerdam ME, Ramage JK, Srirajaskanthan R.Nutrients. 2023 Aug 30;15(17):3787. doi: 10.3390/nu15173787).
Additionally, for insulinomas, which are mostly benign entities, loco regional therapy can work very well in controlling symptoms by offering an organ preserving approach and avoiding surgery. There are many reports on EUS-guided radiofrequency ablation. This also can be added.
opic Ultrasound-Guided Locoregional Treatments for Pancreatic Neuroendocrine Neoplasms.
Masciangelo G, Campana D, Ricci C, Andrini E, Rakichevikj E, Fusaroli P, Lisotti A.Curr Oncol. 2025 Feb 16;32(2):113. doi: 10.3390/curroncol32020113.Author Response
We would like to thank the reviewer for taking the time to evaluate our manuscript and for providing constructive and thoughtful comments. We appreciate the valuable insights and suggestions, which have helped us to significantly improve the quality and clarity of our work. We have carefully addressed each comment in the revised version of the manuscript and provided detailed responses explaining the changes made. We believe these revisions have strengthened the overall contribution of our paper and clarified key points for readers. Thank you again for your time, expertise, and efforts in reviewing our work. Attached you can find the response to your comments.
Comment 1: I believe that a paragraph focused on medical nutrition therapy (mostly parenteral nutrition) could be useful. GNET associated intestinal failure is not rare and can be a major challenge as malnutrition can be very difficult to mitigate in these cases. There are some publications already reporting on this situation.(Feasibility of Home Parenteral Nutrition in Patients with Intestinal Failure Due to Neuroendocrine Tumours: A Systematic Review.Clement DSVM, Brown SE, Naghibi M, Cooper SC, Tesselaar MET, van Leerdam ME, Ramage JK, Srirajaskanthan R.Nutrients. 2023 Aug 30;15(17):3787. doi: 10.3390/nu15173787).
Response 1: Although the topic we are talking about is functioning NENs and intestinal failure is more unfrequent and more frequently seen after surgical resection, in some cases a functional intestinal failure can be seen in carcinoid syndrome tumors and we have added a comment on parenteral nutrition as suggested and added the reference suggested accordingly.
Comment 2: Additionally, for insulinomas, which are mostly benign entities, loco regional therapy can work very well in controlling symptoms by offering an organ preserving approach and avoiding surgery. There are many reports on EUS-guided radiofrequency ablation. This also can be added. Endoscopic Ultrasound-Guided Locoregional Treatments for Pancreatic Neuroendocrine Neoplasms. Masciangelo G, Campana D, Ricci C, Andrini E, Rakichevikj E, Fusaroli P, Lisotti A.Curr Oncol. 2025 Feb 16;32(2):113. doi: 10.3390/curroncol32020113.
Response 2: we have added the coment and added the reference accordingly. Thank you!
Reviewer 2 Report
Comments and Suggestions for Authors
Thank you for the opportunity to read the article “Nutritional Management of Functioning GEP-NENs” which I found to be both interesting and informative.
The issue of malnutrition in oncology patients is already well recognized and clinically significant. In the case of neuroendocrine tumors (NETs), it is further complicated by the hormonal activity of the tumors. In this context, the presented review is valuable and may contribute to improved care for patients with functioning NETs. The article is generally well-written.
I would like to offer a few minor suggestions for improvement:
- Could the authors consider including a brief definition of malnutrition?
- It might also be helpful to dedicate a short paragraph to general recommendations for the management of malnutrition in oncology patients, including strategies to increase dietary caloric intake and a brief overview of available anabolic agents.
Author Response
We would like to thank the reviewer for taking the time to evaluate our manuscript and for providing constructive and thoughtful comments. We appreciate the valuable insights and suggestions, which have helped us to significantly improve the quality and clarity of our work. We have carefully addressed each comment in the revised version of the manuscript and provided detailed responses explaining the changes made. We believe these revisions have strengthened the overall contribution of our paper and clarified key points for readers. Thank you again for your time, expertise, and efforts in reviewing our work. Attached you can find the response to your comments.
Comment 1:
- Could the authors consider including a brief definition of malnutrition?
Response 1: We have added GLIM criteria for definition of malnutrition in the introduction accordingly.
Comment 2:
- It might also be helpful to dedicate a short paragraph to general recommendations for the management of malnutrition in oncology patients, including strategies to increase dietary caloric intake and a brief overview of available anabolic agents.
Response 2: we have added in the introduction a short paragraph of ESPEN recommendations in oncology patiets and a comment on anabolic agents